# Cell-Free Supernatant of *Lactiplantibacillus plantarum* 90: A Clean Label Strategy to Improve the Shelf Life of Ground Beef Gel and Its Bacteriostatic Mechanism

**DOI:** 10.3390/foods12224053

**Published:** 2023-11-07

**Authors:** Jing Wang, Lilan Xu, Luping Gu, Yuanqi Lv, Junhua Li, Yanjun Yang, Xiangren Meng

**Affiliations:** 1College of Tourism and Culinary Science, Yangzhou University, Yangzhou 225127, China; 008159@yzu.edu.cn; 2Jiangxi Key Laboratory of Natural Products and Functional Food, Jiangxi Agricultural University, Nanchang 330045, China; xulilan@jxau.edu.cn; 3School of Food Science and Technology, Jiangnan University, Wuxi 214122, China; guluping@jiangnan.edu.cn (L.G.); lvyuanqi@haut.edu.cn (Y.L.); lijunhua@jiangnan.edu.cn (J.L.); 4Chinese Cuisine Promotion and Research Base, Yangzhou University, Yangzhou 225127, China

**Keywords:** *Lactiplantibacillus plantarum* 90, cell-free supernatant, clean label, ground beef preservation, bacteriostatic mechanism

## Abstract

Lactic acid bacteria metabolites can be used as a clean-label strategy for meat products due to their “natural” and antibacterial properties. In this study, the feasibility of using cell-free supernatant of *Lactiplantibacillus plantarum* 90 (LCFS) as a natural antibacterial agent in ground beef was investigated. The sensitivity of LCFS to pH, heat and protease, as well as the changes of enzyme activities of alkaline phosphatase (AKP) and Na^+^/K^+^-ATP together with the morphology of indicator bacteria after LCFS treatment, were analyzed to further explore the antibacterial mechanism of LCFS. The results showed that the addition of 0.5% LCFS inhibited the growth of microorganisms in the ground beef gel and extended its shelf-life without affecting the pH, cooking loss, color and texture characteristics of the product. In addition, the antibacterial effect of LCFS was the result of the interaction of organic acids and protein antibacterial substances in destroying cell structures (cell membrane, etc.) to achieve the purpose of bacteriostasis. This study provides a theoretical basis for the application of LCFS in meat products and a new clean-label strategy for the food industry.

## 1. Introduction

As a nutrient-rich matrix, meat and its products play an important role in the human diet. However, the abundance of nutrients also makes it a natural medium for both spoilage and pathogenic microorganisms [1]. Therefore, in the processing of such products, it is often necessary to add preservatives to ensure their safety, as well as texturizers to improve their sensory quality [2,3]. Nevertheless, in recent years, pursuing a “natural” diet via “clean label” products, has become a consumer market trend. The characteristics of meat products leave the meat industry facing a major challenge in developing clean-label meat products [4]. To this end, researchers have carried out relevant study [4] to make meat products relatively more “natural, healthy and safe”. The clean-label alternatives for current common additives in meat processing are shown in Figure 1. It can be seen from Figure 1 that most of the clean-label substitutes for additives in meat products are natural substances of plant origin. Among them, clean-label alternatives for antimicrobial agents in meat include lactoferrin and lysozyme from animals [5,6] as well as lactic acid bacteria and their metabolites [7]. However, alternatives of animal origin are relatively expensive, while some plant-derived substances can negatively affect the aroma of the final products [4]. In the context of natural antimicrobials being favored, lactic acid bacteria and their metabolites have attracted widespread attention from consumers.

Nowadays, the antibacterial effects of many lactic acid bacteria and their metabolites have been confirmed, and they have been “generally recognized as safe” (GRAS) by relevant regulatory agencies and approved for use in the field of food antisepsis [7], and most probiotics have bacteriostatic effects due to the amounts and diversity of metabolites present in their cell-free supernatants (CFS) [8,9]. Moreover, compared with live probiotics, CFS are more stable in food systems [10]. The application of CFS helps to reduce the use of chemical preservatives, which meets the consumer demand for “fresh, natural” food [11]. Therefore, the addition of CFS may be an effective clean-label strategy in the field of the preservation of meat products.

*Lactobacillus plantarum* (*L. plantarum*) is one of the most widespread species of the genus *Lactobacillus*, and is widely used in the food industry [8]. It has been reported by Arrioja-Bretón et al. [12] that CFS of *L. plantarum* was effective in reducing the microbial load of inoculated bacteria in beef, mainly *S. Typhimurium* and *L. monocytogenes*. CFS marinade had a certain effect on the color of raw beef, while on grilled meat, changes were scarcely detected. Houssam Abouloifa et al. [13] studied the bio-preservation effect of probiotic *Lactiplantibacillus plantarum* S61 against *Rhodotorula glutinis* and *Listeria monocytogenes* in poultry meat, and the results showed that the *L. plantarum* S61 strain allowed the reduction of *L. monocytogenes* in minced poultry meat during 7 days of storage at 4 °C and also improved the physicochemical and color parameters of poultry minced meat. In addition, in our previous study [14], it was found that the CFS of *L. plantarum* 90 (LCFS) had the best inhibitory effect on typical spoilage bacteria in liquid egg among several CFSs of lactic acid bacteria. However, considering the differences in food systems, its preservation effect on ground beef products remains to be explored. Meanwhile, the antimicrobial substances in LCFS and their antimicrobial mechanisms remain unclear. Some studies [15,16,17] have shown that the antibacterial effect of cell-free supernatant of *L. plantarum* was related to the organic acids and bacteriocins produced by it, and the antibacterial effect was achieved by destruction of the cytoplasm, cell membrane, and cell wall of target microorganisms. However, the antibacterial substances produced by different strains and their antibacterial mechanisms are specific, and no study has been found on the antibacterial substances of *L. plantarum* 90 and their mechanisms of action. Therefore, this study aims to investigate the feasibility of LCFS as a clean-label strategy for the preservation of ground beef products, and further explore the antibacterial substances in LCFS and its antibacterial mechanisms, so as to provide a theoretical basis for its application as a clean-label alternative in the future.

## 2. Materials and Methods

### 2.1. Chemicals and Materials

Beef and butter were purchased from a local supermarket (Yangzhou, Jiangsu, China). All the microbial media used in the experiments were bought from Qingdao Hope Bio-Technology Co., Ltd. (Qingdao, Shandong, China). AKP and Na^+^/K^+^-ATP kits were obtained from Nanjing Jiancheng Bioengineering Institute (Nanjing, China). Neutral protease (2 × 10^5^ U/g), papain (8 × 10^5^ U/g), pepsin (1200 U/g) and trypsin (2.29 × 10^6^ U/g) were purchased from Sinopharmate Chemical Reagent Co., Ltd. (Shanghai, China); proteinase K (3 × 10^5^ U/g) was purchased from Solaibao Biotechnology Co., Ltd. (Guangzhou, China); Oxford cups (8 × 6 × 10 mm) were bought from Shanghai Jingchong Electronic Technology Development Co., Ltd. (Shanghai, China); 96-well plates were purchased from Hengkang Medical Group Inc. (Wuxi, China). All the other reagents were of analytical grade.

### 2.2. Bacteria and Culture Conditions

The strain of *L. plantarum* 90 was obtained from Weikang Biotechnology Co., Ltd. (Suzhou, Jiangsu, China). The strains of *Pseudomonas putida* I3, *Microbacterium* sp. ALBL_062, *Salmonella* CICC 21513 and *Staphylococcus aureus* ATCC 6538 came from our own laboratory. The strain of *L. plantarum* 90 (1%, *v*/*v*) was inoculated into MRS media aerobically cultivated at 37 °C for 24 h. The strains of *Pseudomonas putida* I3 and *Microbacterium* sp. ALBL_062 (1%, *v*/*v*) were inoculated into LB media and aerobically cultivated at 30 °C for 24 h. The strains of *Salmonella* CICC 21513 and *Staphylococcus aureus* ATCC 6538 (1%, *v*/*v*) were inoculated into LB media and aerobically cultivated at 37 °C for 24 h. Before each experiment, frozen strains were twice cultivated.

### 2.3. Preparation of LCFS

*L. plantarum* 90 (1%, *v*/*v*) was cultured in MRS media and incubated at 37 °C for 18 h. Bacteria and supernatant were separated by centrifugation (8000× *g*, 10 min). Then, the supernatant was filtrated using 0.22 µm sterile filters. The filtered supernatant was concentrated by a rotary evaporator and pre-frozen in a −80 °C refrigerator, after which it was freeze-dried in a lyophilizer (SCIENTZ-10N, Ningbo Xinzhi Biotechnology Co., Ltd., Ningbo, China) for 48 h. The freeze-dried LCFS were collected in sterile sampling bags for later use.

### 2.4. Preservation of Ground Beef Gel

#### 2.4.1. Sample Preparation

The beef and butter were chopped into small pieces using a meat grinder and then mixed to a fat-to-lean ratio of 4:6. Subsequently, 1% NaCl, 20% water and different contents of LCFS (0%, 0.5%, 1%, 2.5%, 5%) were added to the mixed system (LCFS were dissolved in water before added to the system). The ground beef mixtures with different LCFS contents were separately stirred clockwise for 5 min. After that, each 30 g of them was loaded into a 50 mL flat-bottom centrifuge tube. Samples in the centrifuge tube were compressed and then heated in a water bath at 100 °C for 30 min to obtain the ground beef gel. Then, the prepared ground beef gel was cut into 1 cm-high cylinders and separately loaded into sterile bags, all samples were stored in the refrigerator at 4 °C for 10 days.

#### 2.4.2. Determination of Microbial Counts of Ground Beef Gel

Samples with different LCFS contents were removed from the refrigerator each day to determine their total aerobic mesophilic count (TAM). The TAM was detected on the basis of our previous study [18] by plate-counting method and the incubation condition was 30 °C, 3 days, aerobic.

#### 2.4.3. Determination of the pH, Cooking Loss and Color of Ground Beef Gel

The pH value of samples were determined according to the method reported by Lu et al. [19] with slight modifications. Ten-gram minced samples of ground beef gel with 9 mL of distilled water were shaken for 30 s and then macerated for 30 min, after which the pH of the supernatant was determined using a pH meter.

Samples of ground beef were weighed before and after cooking (cooled to approximately 25 °C), and the cooking loss was calculated according to Equation (1):Cooking loss (%) = (M_0_−M_1_)/M_0_ × 100(1)
where M_0_ and M_1_ represent the weight of the ground beef samples before and after cooking, respectively.

The L*, a* and b* values of ground beef gels were determined by Ultra Scan Pro1 166 spectrometer (Hunter Lab, Reston, VA, USA) against standard black and white background. (where L* = lightness on a 0–100 scale from black to white, a* = scale of red (+) or green (−), b* = scale of yellow (+) or blue (−)). The gel samples were sealed in a plastic bag, three points of the bag were analyzed and an average value was obtained for each sample. The ΔE is the total color change of ground beef gels with different LCFS additions calculated from Equation (2) [20]:(2)$$\Delta E=\sqrt{{\left(dL*\right)}^{2}+{\left(da*\right)}^{2}+{\left(db*\right)}^{2}}$$

#### 2.4.4. Determination of the Texture of Ground Beef Gel

The texture property was determined according to our previous method [20]. The gels that had been refrigerated overnight were removed from the fridge. Before measurement, the gels were brought to room temperature (25 °C). The compression test of the gel samples was carried out using a P/35 probe integrated with a texture analyzer (Stable Micro Systems, Godalming, UK). Typical parameters were as follows. Operating mode: texture profile analysis (TPA), pre-test speed: 2.0 mm/s, test speed: 2.0 mm/s, post-test speed: 2.0 mm/s; trigger mode: strain, strain: 50%; trigger type: auto (force); trigger force: 5.0 g.

### 2.5. Antimicrobial Properties of LCFS

#### 2.5.1. Dynamics of Growth and Bacteriostatic Property

The secondary activated strain of *L. plantarum* 90 was inoculated into the MRS liquid medium with 1% (*v*/*v*) of the inoculation amount and cultivated at 37 °C for 24 h. In the course of cultivation, samples of bacteria suspension were taken every 2 h, and the OD600 values of the samples were measured. After that, the bacterial suspension was centrifuged at 8000× *g* for 10 min, and the supernatant was taken to obtain LCFS. The LCFS was filtered with a 0.22 µm sterile filter for reserve. The pH of LCFS was determined using a digital pH meter (Delta320, Mettler-Toledo Instrument Co., Ltd., Shanghai, China). Then, the antibacterial activity of LCFS was determined by Oxford cup method [21] with *Pseudomonas putida* I3 as the indicator strain.

#### 2.5.2. MIC and MBC of LCFS

The freeze-dried LCFS was resuspended in sterile LB broth to reach an initial concentration of 2 g/mL, after that it was diluted to a concentration gradient of 20, 18, 16, 14, 12, 10, 8 mg/mL. A total of 100 µL different concentrations of LCFS were mixed with 100 μL indicator bacteria (approximately 10^7^ CFU/mL, *Pseudomonas putida* I3, *Microbacterium* sp. ALBL_062, *Salmonella* CICC 21513 and *Staphylococcus aureus* ATCC 6538) in microporous plates, and 100 µL sterile LB broth mixed with 100 µL indicator bacteria were set as the control group. Meanwhile, to ensure the sterility of LB broth, 200 µL LB broth was used as the blank control. The microporous plates containing different indicator bacteria were cultured at the optimal growth temperature of each indicator bacteria for 24 h, and the turbidity of each hole in the microporous plates was observed. The lowest concentration of LCFS corresponding to the hole without turbidity was the minimum inhibitory concentration (MIC) of LCFS to the indicator bacteria. Then, 50 µL mixture was taken from each well of the microplate and the number of viable bacteria was counted by plate-counting method (PCA; pour plate), while the lowest concentration of LCFS with completely inactivated indicator bacteria was the minimum bactericidal concentration (MBC) of LCFS for each indicator bacteria.

#### 2.5.3. Growth Curve and Time-Killing Curve

The growth curves and bactericidal curves of the four indicators mentioned above were measured by the method of Yi et al. [22]. Briefly speaking, the indicator bacteria (at the logarithmic stage of growth) were exposed to LCFS at the concentrations of 1 × MIC or 2 × MIC, respectively. After that, they were cultured at their own optimum growth temperature for 24 h, and samples were taken at 2 h intervals (i.e., 0, 2, 4, 6, 8, 10, 12, 14, 16, 18, 20 and 24 h), while the OD600 of each sample was determined immediately. The determination method of the time-killing curves was similar to that of the growth curves; that is, the indicator bacteria were exposed to LCFS of 1 × MIC or 2 × MIC, cultured at the optimum temperature for 10 h for each bacterium, and the number of viable bacteria in each sample was calculated by plate-counting method for 0, 2, 4, 6, 8, and 10 h, respectively. Moreover, in both of the experiments, untreated samples were used as controls.

#### 2.5.4. Effects of pH, Heat and Enzymes on LCFS

The tolerance of LCFS to different pH values, different heat treatment and different enzymes was determined by referring to the method of Peng et al. [23]. In order to evaluate the stability of LCFS to pH, its pH was adjusted to 2.0–9.0 with 3 M sodium hydroxide or 3 M hydrochloric acid, and a sample of pH 3.62 (the original pH of LCFS) was used as a control. To evaluate the thermal stability of LCFS, it was heated at 60 °C for 10/30 min, 80 °C for 10/30 min, and 100 °C for 10/30 min in a thermostatic water bath and at 121 °C for 15 min in an autoclave. In order to evaluate the enzyme sensitivity of LCFS, it was treated with 1 mg/mL neutral proteinase, pepsin, trypsin, papain and proteinase K. Meanwhile, the untreated LCFS was used as control. Finally, all samples were incubated in a water bath at 37 °C for 4 h and then enzyme-inactivated at 100 °C for 5 min. After pH, heat and enzyme treatment, the bacteriostasis of the treated LCFS was measured by the Oxford cup method with the four strains listed in Section 2.2 as indicators.

#### 2.5.5. Enzyme Activities of AKP and Na^+^/K^+^-ATP from Indicator Bacteria

The enzyme activities of AKP and Na^+^/K^+^-ATP from indicator bacteria were measured according to the method described by Sun et al. [24] with a slight change. The indicator bacteria cultured to the logarithmic growth stage were washed and re-suspended with sterile water; LCFS was then added to make the concentration reach MIC, while the control group was added with an equal volume of sterile water. All samples were then cultured at their optimum growth temperature for 5 h, and AKP and Na^+^/K^+^-ATP activities in the supernatant of indicator bacteria were measured by AKP and Na^+^/K^+^-ATP detection kits, respectively.

#### 2.5.6. Scanning Electron Microscopy (SEM)

The indicator bacteria growing to logarithmic stage were treated with LCFS of 1 × MIC for 5 h while untreated indicator bacteria were used as controls. After that, the indicator bacteria were centrifuged at 1000× *g* for 5 min at 4 °C. Then, the cells were collected and washed with sterile water and fixed with 2.5% glutaraldehyde (0.1 mol/L phosphate buffer, pH 7) overnight. Finally, the cells’ pretreatment was carried out by the method described by Lee et al. [25]. That is, samples were successively eluted with a gradient of low- to high-concentration alcohol and dried with a CO_2_ critical point dryer. Afterwards, the fixed samples were sputter-coated with gold and observed in a scanning electron microscope (ESEM, Quanta-200F, FEI, Hillsboro, OR, USA) at an acceleration voltage of 3 kV.

### 2.6. Statistical Analysis

Statistical analyses of data were performed using the SPSS 16.0 software program. Analysis of variance (ANOVA) was applied and *p* < 0.05 was considered statistically significant. Duncan and least significant differences (LSD) tests were used to measure the significance among the tested parameters. The results are reported as the mean and standard deviation of these measurements. All the diagrams were plotted using Origin 9 64 Bit software.

## 3. Results and Discussions

### 3.1. Preservation of Ground Beef Gel

#### 3.1.1. Effect of LCFS on TAM of Ground Beef Gels during Cold Storage

It is particularly important to evaluate the antibacterial properties of natural antibacterial substances in food systems rather than in vitro, because they may interact with food components and affect their antibacterial properties [4]. Due to the differences in food systems, the antibacterial effects of the same substance may also be quite different. To investigate the effect of LCFS on the preservation of ground beef gel, we examined TAM changes of samples with different amounts of LCFS during cold storage. The results are shown in Figure 2.

As shown in Figure 2, there was little difference in the initial colony number at 0 d among all samples, and the TAM of each sample gradually increased with the extension of cold storage. It is noteworthy that the TAM of the samples supplemented with LCFS was lower than that of the samples without LCFS throughout the storage experiment, which indicated that the addition of LCFS was able to inhibit the growth of microorganisms in the ground beef gel, and was also preliminary verification of the feasibility of its application as a clean-label strategy in ground beef products. Similarly, a previous study by Houssam Abouloifa et al. [13] showed that *Lpb. plantarum* S61 strain allowed the reduction of *L. monocytogenes* in minced poultry meat during 7 days of storage at 4 °C. In our experiment, the TAM of the samples without LCFS exceeded the limit of microbiological acceptable levels (4 log _10_ CFU mL^−1^) on the sixth day of refrigeration, while the TAM of the samples with 0.5% and 1% LCFS exceeded this limit on the eighth day of refrigeration, and the TAM of the samples with 2.5% and 5% LCFS did not exceed this limit during the storage period. These results also indicated that the inhibition of microorganisms in ground beef gel by LCFS was dose-dependent.

#### 3.1.2. Effect of LCFS on pH, Cooking Loss and Color of Ground Beef Gels

The pH value is one of the important indexes that affect the quality of ground beef gel products. As is shown in Table 1, with the increase of LCFS addition, the pH value of the ground beef gel decreased significantly. This is because lactic acid bacteria was metabolized to produce organic acids during growth [23], making the cell-free supernatant acidic, and the addition of acidic LCFS reduced the pH of the system. The organic acids in LCFS can inhibit the growth of microorganisms in the ground beef gel, which may be the reason why the addition of LCFS extended the shelf life of the samples to a certain extent.

Cooking loss directly affects product yield, while color and texture affect the acceptance and thus are essential for ground beef products. As can be seen in Table 1, there was no significant change in the cooking loss of ground beef gel when the addition of LCFS was 0.5% and 1% compared to the samples without LCFS. The cooking loss of the ground beef gel was significantly reduced when the amount of LCFS reached 2.5% and 5%. Combined with the texture characteristics shown in Table 2, it is analyzed that this might be due to the acidic pH environment (pH < 5) that led to premature denaturation and aggregation of myofibrillar protein, resulting in looser heat-induced gel particles formed in the later stage [26]. Loose gels with larger pores may have trapped more free water, thus reducing the cooking losses of the ground beef gel.

As for the color of each sample, the results showed that, similar to the pattern of cooking losses, there was no significant change in the color of ground beef gel when the addition of LCFS was 0.5% and 1%. It may be that subtle color differences were not apparent after the samples were cooked. This was similar to the previous findings of Arrioja-Bretón et al. [12]; that is, CFS of *Lb. plantarum* NRRL B-4496 marinade had a certain effect on the color of raw beef, while the color changes were scarcely detected on the grilled meat. When the LCFS supplementation reached 2.5% and 5%, the acidic environment caused partial degeneration of myofibril and affected the color of the ground beef gel, which was manifested by the reduction of L* and a* values. It has been shown that ground beef has a larger L* value in a neutral environment and a lower L* value in an acidic environment, while the lower content of oxygenated myoglobin, which provided red color, led to its lower a* value in a low pH environment [27].

#### 3.1.3. Effect of LCFS on the Texture Properties of Ground Beef Gels

Texture property is an important index to evaluate the quality of minced meat products, so the effects of LCFS addition on the texture properties of ground beef gel are shown in Table 2. The results showed that the addition of 0.5% LCFS had no significant effect on the texture of ground beef gel (the TPA parameters of the 0.5% LCFS group were not significantly different from that of the control group (*p* < 0.05), and when the addition of LCFS exceeded 0.5% (1%, 2.5%, 5%), the texture indexes (particularly hardness, springiness, adhesiveness and chewiness) of minced beef gel significantly decreased (*p* < 0.05) with the increase of LCFS. The effect of LCFS addition on cohesiveness was not so sensitive. When the amount of LCFS addition was greater than 1%, the difference appeared, and the overall trend was to first decrease (2.5%) and then increase (5%). Combined with the pH values in Table 1, it is not difficult to understand that this was caused by the reduction of pH of the ground beef gel. The process of heat-induced gel formation of ground beef was accompanied by the structure of myofibrillar protein unfolding and then aggregation [28], because pH changes the charge distribution and thus affects the protein structure. As mentioned above, acidic pH caused partial premature denaturation and aggregation of myofibrillar protein, thus the gel structure was looser and coarser [26].

In conclusion, the addition of appropriate quantities of LCFS can inhibit the growth of microorganisms in ground beef gel and prolong the shelf life of the product without affecting its quality, and can be used as an alternative additive that fits the concept of “clean label”.

### 3.2. Antimicrobial Properties of LCFS

#### 3.2.1. Dynamics of Growth and Bacteriostatic Property

The growth curve of *L. plantarum* 90 in MRS medium, the change of pH and the bacteriostasis of its CFS (with *Pseudomonas putida* I3 as the indicator bacteria) are shown in Figure 3. As it can be seen from Figure 3, *L. plantarum* 90 grew rapidly in MRS medium and entered a logarithmic growth phase within 2 h, while it gradually stabilized at 10 h until the end of culture. At the same time, along with the growth of *L. plantarum* 90, the pH value of LCFS gradually decreased; specifically, the pH value decreased sharply in the logarithmic growth stage, while it gradually stabilized in the stable growth stage. In addition, the LCFS began to show antibacterial activity at 6 h, and reached its maximum antibacterial activity at 18 h. Therefore, LCFS cultured for 18 h was used in the subsequent experiments.

It has been reported [29] that lactic acid bacteria can convert glucose into organic acids (mainly lactic acid) during their growth, thereby reducing the pH value of the system, while organic acids can inhibit the growth of target microorganisms by changing the osmotic pressure of microorganisms, increasing the permeability of cell membranes, and inhibiting the synthesis of relevant DNA and proteins. In addition, as secondary metabolites, bacteriocins generally start to be produced in the middle-late logarithmic period of bacterial growth and reach a maximum value during the stable period at which the growth rate of the strain gradually decreases [30]. Bacteriocins generally have membrane activity, which can destroy the permeability of cell membranes and lead to the leakage of cell contents, thus achieving antibacterial effects [31]. Combining the experimental results in Figure 3, the antimicrobial activity of LCFS did not always increase as the pH decreased (after 18 h, the pH continued to decrease and the antimicrobial activity did not increase, but decreased). It was initially speculated that the bacteriostatic effect of LCFS might be caused by the combined action of organic acid and other bacteriostatic substances (maybe bacteriocin) in it, because there is precedent [16,17,32] for the isolation of bacteriocins from lactic acid bacteria, especially *Lactobacillus plantarum*.

#### 3.2.2. MIC and MBC of LCFS

In order to further understand the antibacterial effect of LCFS and explore its antibacterial characteristics, the MIC and MBC of LCFS against four indicator bacteria (two spoilage bacteria and two pathogenic bacteria; *Pseudomonas putida* I3 and *Salmonella* CICC 21513 are Gram-negative bacteria, while *Microbacterium* sp. ALBL_062 and *Staphylococcus aureus* ATCC 6538 are Gram-positive bacteria) were determined, and the results are shown in Table 3. That is, the MIC of LCFS for all four indicator bacteria was 8 mg/mL, while MBC was 16 mg/mL. Wang et al. [15] reported previously that the MIC of the CFS of *L. plantarum* (Accession: MW282954) on *Proteus mirabilis* was 12.5 mg/mL. In contrast, CFS of *L. plantarum* 90 has a much smaller MIC.

It can be seen from Table 3 that LCFS had an inhibitory effect on both Gram-positive and Gram-negative bacteria. However, most bacteriocins (such as nisin) have antibacterial activity only against Gram-positive bacteria and do not inhibit the growth of Gram-negative bacteria. Because most bacteriocins are cationic peptides, they cannot cross the outer membrane of Gram-negative bacteria, where the cations in core components such as lipopolysaccharides and phospholipid bilayers stabilize the outer membrane by non-covalent binding [33]. However, the CFS of many lactic acid bacteria contain lactic acid, which acts as a permeability agent for the outer membrane of Gram-negative bacteria and can release the lipopolysaccharide in their outer membranes. Lactic acid and bacteriocin action at the same time can form a fence effect, so as to better inhibit the growth of various spoilage and pathogenic bacteria.

#### 3.2.3. Growth Curve and Time-Killing Curve

The growth curves of four indicator bacteria in the presence of LCFS (1 × MIC, 2 × MIC) are shown in Figure 4a. As can be seen from Figure 4a, compared with the control group, the growth of bacteria in the group with LCFS at 2 × MIC concentration was completely inhibited within 24 h. Under LCFS of 1 × MIC, the growth of *Salmonella* CICC 21513 was still inhibited within 24 h, while *Pseudomonas putida* I3, *Microbacterium* sp. ALBL_062 and *Staphylococcus aureus* ATCC 6538 began to grow at 22 h, 20 h and 22 h, respectively. This also indicated that the inhibition effect of LCFS on different strains was different, which may be related to the cell structure of each indicator bacteria, culture temperature and the antibacterial mechanism of LCFS.

Figure 4b shows the time-killing curve of LCFS against four indicator bacteria over 10 h. As shown in Figure 4b, LCFS of 1 × MIC could inhibit the growth of *Pseudomonas putida* I3 and *Microbacterium* sp. ALBL_062. When the concentration of LCFS reached MBC (2 × MIC), the counts of these two bacteria decreased steadily over time from 2 h (*Pseudomonas putida* I3 decreased from 7.39 log_10_ CFU mL^−1^ to 2.77 log_10_ CFU mL^−1^, while *Microbacterium* sp. ALBL_062 decreased from 7.49 log_10_ CFU mL^−1^ to 2.95 log_10_ CFU mL^−1^). As for the two pathogens, LCFS of 1 × MIC made the count of *Salmonella* CICC 21513 decrease slowly from 6.77 log_10_ CFU mL^−1^ to 4.00 log_10_ CFU mL^−1^, while LCFS of MBC made the count of *Salmonella* CICC 21513 decrease sharply from 6.68 log_10_ CFU mL^−1^ to 3.80 log_10_ CFU mL^−1^ within 2 h. Similarly, under LCFS of 1 × MIC, the count of *Staphylococcus aureus* ATCC 6538 declined tardily from the initial 6.83 log_10_ CFU mL^−1^ to the final 4.77 log_10_ CFU mL^−1^. When the amount of LCFS reached MBC, the count of *Staphylococcus aureus* ATCC 6538 dropped rapidly from 6.53 log_10_ CFU mL^−1^ to 2.77 log_10_ CFU mL^−1^ within 2 h and then leveled off. In general, *Salmonella* CICC 21513 and *Staphylococcus aureus* ATCC 6538 were more sensitive to LCFS than *Pseudomonas putida* I3 and *Microbacterium* sp. ALBL_062.

#### 3.2.4. Effects of pH, Heat and Enzymes on LCFS

In order to further explore the antibacterial properties of LCFS, the effects of pH, enzyme and heat treatment on the antibacterial effect of LCFS were studied, and the results are shown in Table 4. It can be seen from the table that pH had a great influence on the bacteriostasis of LCFS. In the range of pH 3~5, the bacteriostatic activity of LCFS gradually weakened with the increase of pH, and the inhibition zone disappeared when pH reached 6 and higher, which was similar to the previous findings of Arrioja-Bretón et al. [12]. These results suggest that the antibacterial effect of LCFS may be mainly derived from the organic acids produced by the metabolism of *L. plantarum* 90. In contrast, heat treatment had little effect on the antibacterial properties of LCFS. Under different heat treatment conditions, LCFS showed good antibacterial properties, indicating that the antibacterial substances in LCFS had good heat resistance. As for the effects of enzyme treatment, it can be seen that the bacteriostasis of LCFS showed a decreasing trend after different enzyme treatments. In particular, after treatment with proteinase K, the antibacterial activity of LCFS against the four indicator bacteria was significantly decreased, indicating that there were still protein-based antibacterial substances (possibly bacteriocins, etc.) in LCFS. In addition, it has been reported [34] that due to the limited content of this substance in CFS, its antibacterial effect was not obvious, and its antibacterial effect could be greatly improved after concentration. In summary, the antibacterial effect of LCFS mainly came from the organic acids in LCFS. Secondly, there were also protein-based bacteriostatic substances in LCFS, and such substances were likely to have better bacteriostatic effects under acidic conditions.

#### 3.2.5. Enzyme Activities of AKP and Na^+^/K^+^-ATP

AKP is a key enzyme attached inside the cell membrane which affects the cell wall synthesis of bacteria [35]. As illustrated in Figure 5a, after LCFS treatment, the ATPase activity in the supernatant of the four indicator bacteria was significantly increased compared with that in the untreated group. These results indicate that LCFS treatment damaged the cell wall of indicator bacteria and led to the increased membrane permeability. This was also consistent with the results of previous studies reported by Lin et al. [36].

Na^+^/K^+^-ATP enzyme exists in bacterial cell membranes, and not only plays an important role in controlling the transmembrane transport of Na^+^/K^+^ and maintaining membrane potential, but also drives the transport of amino acids, sugars and other substances in cells, helping to maintain normal physiological metabolism of bacterial cells [37]. Bacterial cells can maintain intracellular and intracellular electrochemical balance by increasing the activity of ATPase, so as to resist the influence of antibacterial agents [38]. The effects of LCFS on the ATPase activity of the four indicator bacteria are shown in the Figure 5b. Consistent with the activity of AKP enzyme, the activity of ATPase in the supernatant of indicator bacteria after LCFS treatment was significantly increased compared with that in the untreated group. These results indicate that LCFS affected cell membrane transport, energy metabolism and other functions of bacteria, thereby inhibiting bacterial growth and eventually leading to bacterial death.

#### 3.2.6. Scanning Electron Microscopy (SEM)

SEM directly reflected the morphological changes of the select bacteria before and after LCFS treatment. The results show that the surface of untreated bacterial cells was smooth, rod-shaped or spherical, with complete structure and uniform size distribution (Figure 6(A1–D1)). However, the morphology and structure of bacteria treated with LCFS for 5 h changed significantly (Figure 6(A2–D2)). Some cell forms were destroyed and holes appeared on the surface, while certain bacteria had uneven surfaces, blurring the boundaries between their cells and sticking on top of each other. Meanwhile, some bacterial cells were obviously depressed and deformed, with large holes and cell rupture, while the overflow of intracellular substances could be clearly observed. A similar result was found in a previous study by Wang et al. [15]; namely, after treatment with the CFS of *L.plantarum*, formation of small holes, stacked and coherent cells, and ruptured cells of *P. mirabilis* were observed. It can be seen that the antibacterial mechanism of LCFS is mainly to destroy the permeability of cell membrane, form pores on the surface of the cell membrane, leading to the leakage of cell contents, and ultimately inhibiting the growth of target microorganisms.

## 4. Conclusions

The addition of 0.5% LCFS could extend the shelf life of ground beef gel to 8 days at 4 °C without affecting the main quality of the product, thus confirming the feasibility of its application as a natural antibacterial agent in meat products. In addition, the antibacterial mechanism of LCFS was mainly to destroy the permeability of the cell membrane, form pores on the surface of the cell membrane and lead to the leakage of cell contents, ultimately inhibiting the growth of target microorganisms. This study provides a theoretical basis for the application of LCFS as a natural antimicrobial agent in meat products as well as a new strategy for developing clean-label meat products. Further work needs to be undertaken to elucidate which component (organic acids, bacteriocins, etc.) contributes to the antibacterial effect of LCFS, and its material basis needs to be further clarified.

## Figures and Tables

**Figure 1 foods-12-04053-f001:**
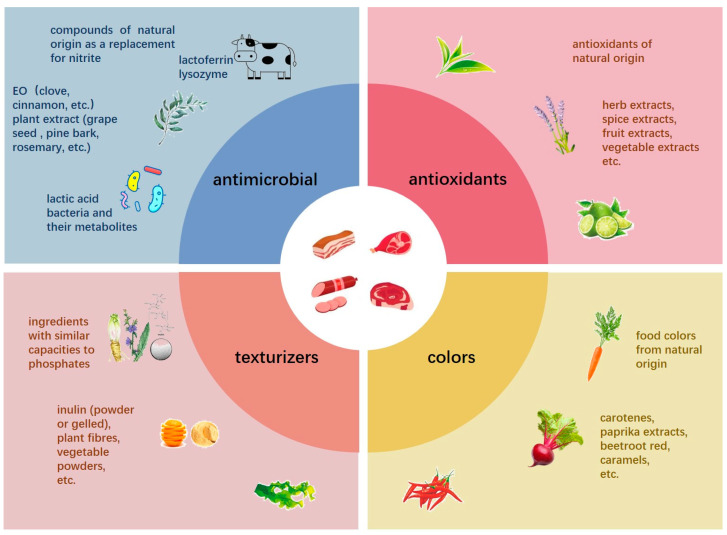
The clean-label alternatives for current common additives in meat processing [4,5,6,7].

**Figure 2 foods-12-04053-f002:**
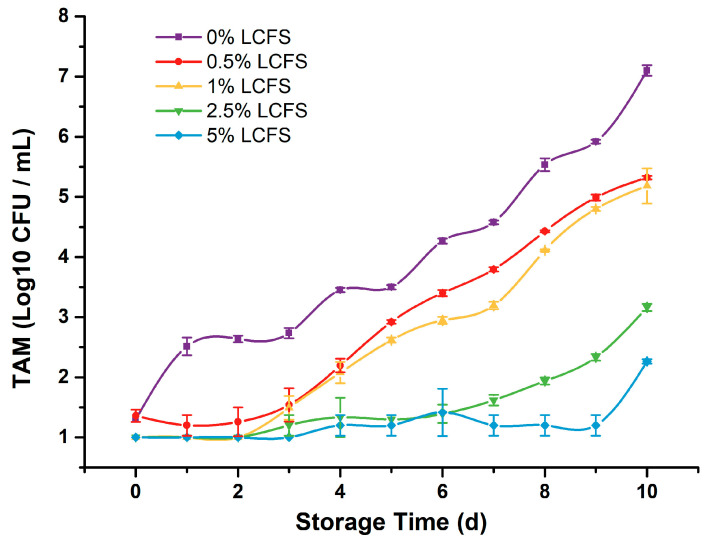
TAM changes of samples with different amounts of LCFS during cold storage.

**Figure 3 foods-12-04053-f003:**
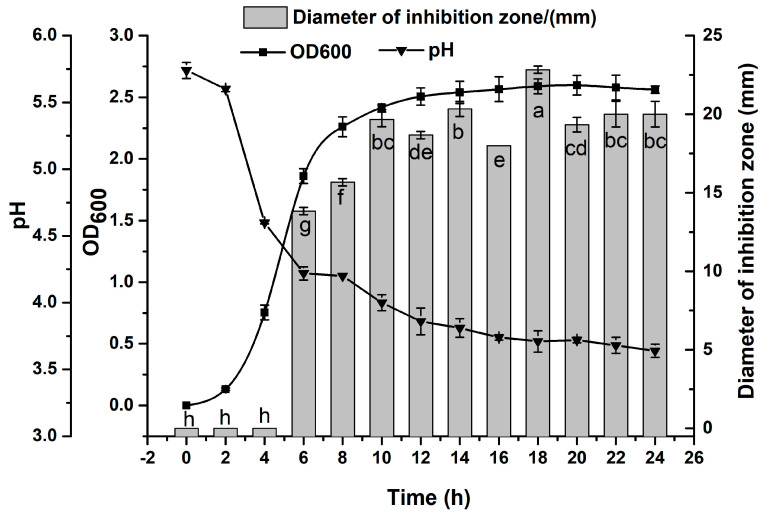
Growth curve and bacteriostatic properties of *L. plantarum* 90. Different superscripts in the bar chart indicate significant differences (*p* < 0.05).

**Figure 4 foods-12-04053-f004:**
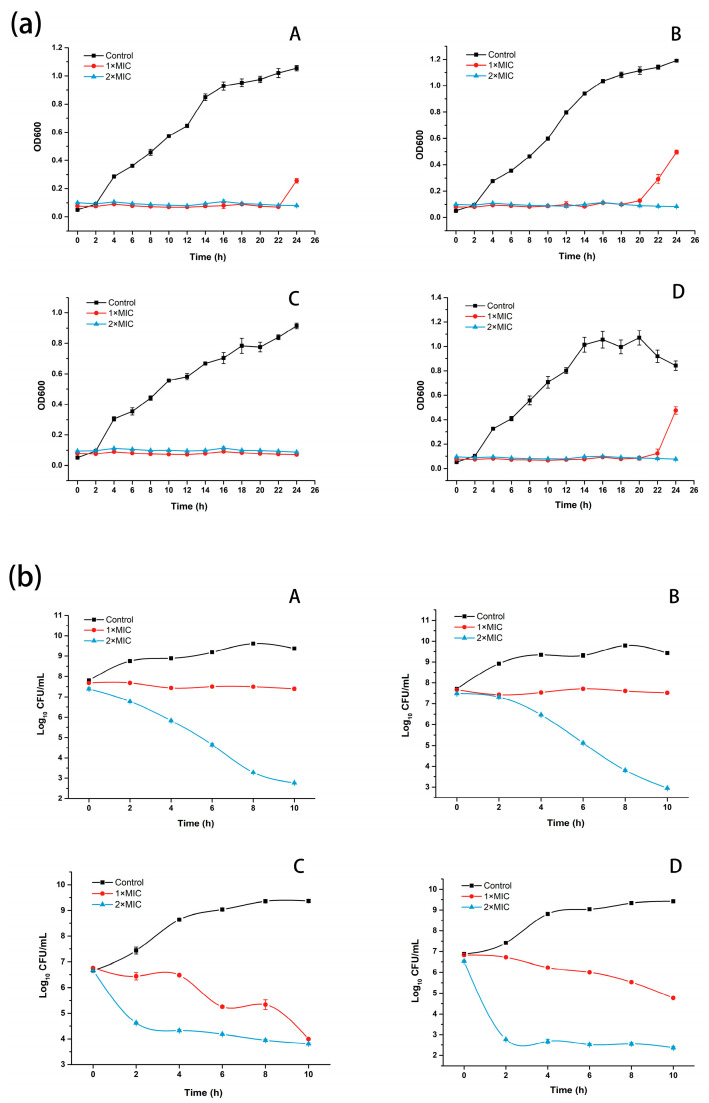
Growth inhibition (**a**) and killing effect (**b**) of LCFS on four indicator bacteria. A, B, C and D represent *Pseudomonas putida* I3, *Microbacterium* sp. ALBL_062, *Salmonella* CICC 21513 and *Staphylococcus aureus* ATCC 6538, respectively.

**Figure 5 foods-12-04053-f005:**
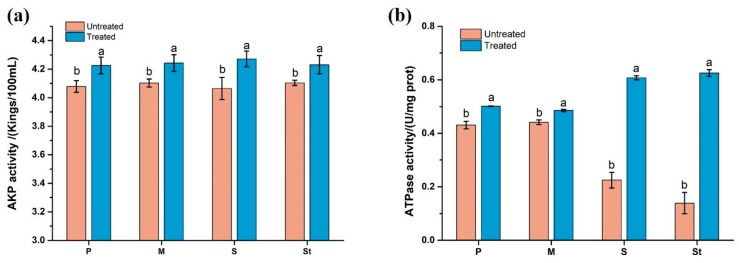
Effects of LCFS treatment on the enzyme of AKP (**a**) and Na+/K+-ATP (**b**) activity of four indicator bacteria. P, M, S and St represent *Pseudomonas putida* I3, *Microbacterium* sp. ALBL_062, *Salmonella* CICC 21513 and *Staphylococcus aureus* ATCC 6538, respectively. Different superscripts in the figure indicate that the same indicator bacteria have significant differences before and after treatment. (*p* < 0.05).

**Figure 6 foods-12-04053-f006:**
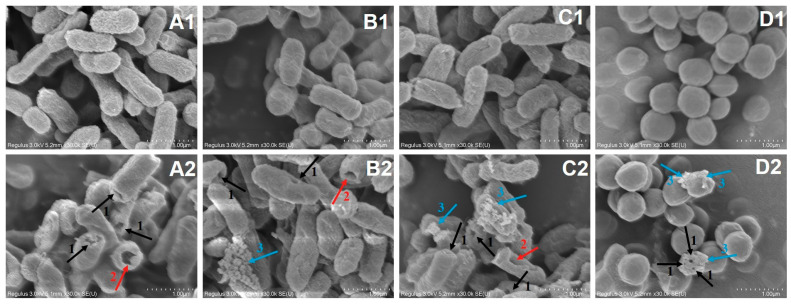
Morphological changes of indicator bacteria before and after LCFS treatment. (**A1**), (**B1**), (**C1**) and (**D1**) represent the morphological characteristics of *Pseudomonas putida* I3, *Microbacterium* sp. ALBL_062, *Salmonella* CICC 21513 and *Staphylococcus aureus* ATCC 6538 before LCFS treatment respectively, while (**A2**), (**B2**), (**C2**) and (**D2**) represented the morphological characteristics of corresponding strains after LCFS treatment. 1, 2, and 3 respectively represent the formation of small pores on the cell surface, the formation of large pores on the cell surface, and cell rupture/intracellular material leakage.

**Table 1 foods-12-04053-t001:** Effects of LCFS on pH, cooking loss and color of ground beef gels.

Content of LCFS/(%)	pH	Cooking Loss (%)	Color			
			L*	a*	b*	ΔE*
0	6.22 ± 0.01 ^a^	51.80 ± 0.42 ^a^	72.64 ± 0.85 ^ab^	1.97 ± 0.18 ^a^	3.66 ± 0.44 ^ab^	8.11 ± 0.96 ^ab^
0.5	5.73 ± 0.04 ^b^	51.72 ± 0.23 ^a^	73.43 ± 0.20 ^a^	1.65 ± 0.15 ^ab^	3.21 ± 0.33 ^b^	9.03 ± 0.13 ^a^
1	5.30 ± 0.02 ^c^	52.64 ± 1.30 ^a^	70.63 ± 2.50 ^bc^	1.81 ± 0.43 ^ab^	4.55 ± 1.33 ^a^	5.96 ± 2.82 ^bc^
2.5	4.71 ± 0.02 ^d^	50.08 ± 0.93 ^b^	73.34 ± 0.50 ^a^	1.40 ± 0.27 ^b^	3.46 ± 0.06 ^ab^	8.89 ± 0.49 ^a^
5	4.36 ± 0.02 ^e^	44.85 ± 0.87 ^c^	68.53 ± 0.32 ^c^	1.85 ± 0.01 ^ab^	4.50 ± 0.72 ^a^	4.06 ± 0.31 ^c^

Mean ± SD with different superscripts within the same column indicates that the values differ significantly (*p* < 0.05). L*, a*, b* and ΔE* represent lightness on a 0–100 scale from black to white, scale of red (+) or green (−), scale of yellow (+) or blue (−) and the total color change, respectively.

**Table 2 foods-12-04053-t002:** Effects of LCFS on the texture properties of ground beef gels.

Content of LCFS (%)	TPA Parameter				
	Hardness (N)	Cohesiveness	Springiness (mm)	Adhesiveness (N)	Chewiness (mj)
0	44.43 ± 1.40 ^a^	0.47 ± 0.03 ^a^	4.76 ± 0.58 ^a^	20.77 ± 0.78 ^a^	99.34 ± 15.57 ^a^
0.5	43.83 ± 1.89 ^a^	0.49 ± 0.03 ^a^	4.19 ± 0.37 ^ab^	21.10 ± 1.64 ^a^	88.55 ± 11.85 ^a^
1	30.30 ± 3.01 ^b^	0.49 ± 0.01 ^a^	4.07 ± 0.28 ^b^	14.80 ± 1.54 ^b^	60.45 ± 10.18 ^b^
2.5	23.46 ± 0.78 ^c^	0.33 ± 0.02 ^c^	3.81 ± 0.08 ^b^	7.67 ± 0.51 ^c^	29.04 ± 1.40 ^c^
5	17.27 ± 1.21 ^d^	0.39 ± 0.06 ^b^	3.73 ± 0.30 ^b^	6.57 ± 0.68 ^d^	24.23 ± 4.33 ^c^

Mean ± SD with different superscripts within the same column indicates that the values differ significantly (*p* < 0.05).

**Table 3 foods-12-04053-t003:** MIC and MBC of LCFS to each indicator bacteria.

Bacterial Pathogen	MIC (mg/mL)	MBC (mg/mL)
*Pseudomonas putida* I3	8	16
*Microbacterium* sp. ALBL_062	8	16
*Salmonella* CICC 21513	8	16
*Staphylococcus aureus* ATCC 6538	8	16

**Table 4 foods-12-04053-t004:** Effects of pH, heat and enzymes on the antibacterial activity of LCFS.

Treatment	Residual Inhibitory Activity (Diameter of Inhibition Zone cm)			
	*Pseudomonas putida* I3	*Microbacterium* sp. ALBL_062	*Salmonella* CICC 21513	*Staphylococcus aureus* ATCC 6538
pH				
3.62 (Control)	2.63 ± 0.05 ^b^	2.25 ± 0.04 ^a^	2.25 ± 0.04 ^a^	2.50 ± 0.07 ^a^
3	2.80 ± 0.04 ^a^	2.17 ± 0.05 ^b^	2.25 ± 0.04 ^a^	2.50 ± 0.08 ^a^
4	2.17 ± 0.09 ^c^	1.97 ± 0.02 ^c^	1.97 ± 0.05 ^b^	2.33 ± 0.05 ^b^
5	1.00 ± 0 04 ^d^	1.08 ± 0.02 ^d^	1.07 ± 0.05 ^c^	0.80 ± 0.00 ^c^
6	0	0	0	0
7	0	0	0	0
8	0	0	0	0
9	0	0	0	0
Heat				
Control	2.68 ± 0.06 ^a^	2.08 ± 0.06 ^ab^	2.03 ± 0.05 ^abc^	2.50 ± 0.09 ^a^
60 °C 10 min	2.65 ± 0.04 ^ab^	2.05 ± 0.04 ^abc^	1.98 ± 0.02 ^bc^	2.37 ± 0.05 ^a^
60 °C 30 min	2.65 ± 0.04 ^ab^	2.03 ± 0.05 ^abc^	2.03 ± 0.02 ^bc^	2.43 ± 0.02 ^a^
80 °C 10 min	2.55 ± 0.04 ^abc^	2.15 ± 0.04 ^a^	2.12 ± 0.06 ^a^	2.38 ± 0.02 ^a^
80 °C 30 min	2.53 ± 0.09 ^bc^	1.82 ± 0.02 ^d^	2.08 ± 0.02 ^ab^	2.38 ± 0.05 ^a^
100 °C 10 min	2.60 ± 0.08 ^abc^	1.95 ± 0.07 ^c^	1.98 ± 0.06 ^bc^	2.47 ± 0.05 ^a^
100 °C 30 min	2.47 ± 0.05 ^c^	2.00 ± 0.08 ^bc^	1.95 ± 0.04 ^c^	2.47 ± 0.06 ^a^
121 °C 15 min	2.58 ± 0.06 ^abc^	2.00 ± 0.00 ^bc^	2.07 ± 0.05 ^ab^	2.48 ± 0.00 ^a^
Enzymes				
Control	2.27 ± 0.02 ^ab^	2.20 ± 0.04 ^a^	1.97 ± 0.05 ^a^	2.48 ± 0.02 ^a^
Neutral protease	2.08 ± 0.02 ^c^	1.73 ± 0.02 ^c^	1.62 ± 0.09 ^cd^	2.40 ± 0.08 ^ab^
Pepsin	2.18 ± 0.02 ^bc^	1.62 ± 0.05 ^c^	1.75 ± 0.04 ^b^	2.42 ± 0.02 ^ab^
Trypsin	2.10 ± 0.04 ^c^	1.88 ± 0.09 ^b^	1.73 ± 0.05 ^bc^	2.33 ± 0.05 ^bc^
Papain	2.23 ± 0.09 ^b^	1.97 ± 0.05 ^b^	1.68 ± 0.02 ^bcd^	2.43 ± 0.05 ^ab^
Proteinase K	1.97 ± 0.05 ^d^	1.67 ± 0.05 ^c^	1.58 ± 0.06 ^d^	2.28 ± 0.02 ^c^

Mean ± SD with different superscripts within the same column indicates that the values differ significantly (*p* < 0.05).

## Data Availability

The data used to support the findings of this study can be made available by the corresponding author upon request.

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
