# Peer review of "Cell-Free Supernatant of Lactiplantibacillus plantarum 90: A Clean Label Strategy to Improve the Shelf Life of Ground Beef Gel and Its Bacteriostatic Mechanism"

_foods, 2023, doi:10.3390/foods12224053_

Round 1
Reviewer 1 Report
Comments and Suggestions for Authors
Please see the attachment

Comments on the Quality of English Language
Minor editing of English language required
Reviewer 2 Report
Comments and Suggestions for Authors
This research deals with a study investigating the feasibility of using LCFS for preserving ground beef products as a clean-label strategy. It also explores the antibacterial substances in LCFS and their antibacterial mechanisms to provide a theoretical basis for its application as a clean-label alternative.
The topic is within the scope of the Journal, the topic is interesting, the manuscript is well-written, and the analyzed parameters are interesting. However, there is a need for some revisions. Please consider the comments below.
The names of microorganisms should be italicized in the text.
Abstract
Line 15. What are these excellent properties?
Line 17. Please delete the word "apply".
Line 18. Please replace "can inhibit" with "inhibited".
The abstract is missing the following information mentioned in the journal's template: (2) Methods: briefly describe the main methods or treatments applied.
Introduction
Lines 33-36. Please split this sentence into two for a better understanding.
Line 38. Please replace the word "while" for a better understanding.
Lines 44-46. Please reformulate this sentence for a better understanding.
Line 51. Please replace the word "while" for a better understanding.
Line 71. Please replace the word "their" before "antibacterial mechanisms" with "its".
Line 78. Please delete the punctuation after "Bioengineering Institute. ".
Line 89. Please replace the word "were" with "was".
Line 98. How was concentrated the lyophilized? What were the lyophilization conditions and the equipment used?
Line 110. "were stored in the refrigerator at 4℃". How long?
Line 114. Please replace the word "while" for a better understanding.
Line 130. What fermentation process?
Line 160. Please replace the word "while" for a better understanding.
Line 164. Please replace the word "while" for a better understanding.
Line 174. Please delete "which" before the "at the concentrations of".
Results and discussions
Line 257. Please replace the word "while" for a better understanding.
Line 260. Please replace "this might because" with "this might be due to".
Line 260. Please add "that" before the word "lead".
Line 262. Please replace the word "while" for a better understanding.
Line 281. Please replace the word "while" for a better understanding.
Table 1. Please delete "/" after "cooking loss" and add "*" after "ΔE". Please mention under the table what are L*, a*, b*, and ΔE*.
Table 2. Please delete "/" after "LCFS".
Line 17, page 9. What are these "relevant macromolecules".
Line 22. Please replace the word "combine" with "Combining".
Figure 3. Please delete "/" after "time" and "diameter of inhibition zone".
Line 34, page 10. Please delete "that of" before "MBC".
Line 35, page 10. Please rephrase "It is not difficult to find that" for a better understanding.
Line 43, page 10. Please replace the word "When" for a better understanding.
Figure 4. Please delete "/" after time. Please replace "Log10 CFU mL-1" with "Log10 CFU/mL".
Line 80, page 11. Please delete the word "intuitively".
Line 82, page 11. Please replace the word "while" for a better understanding.
Figure 5. Please replace "100 mL-1" with " /100 mL".
Line 151. Please replace the word "while" for a better understanding.
A comparison of the results with data from the literature is missing in the results and discussions.
Conclusions
Please reformulate the paragraph from lines 146-153, which is repetitive in the manuscript since it appears in the introduction, results & discussion sections. Please emphasize the implications of the findings, explaining how the work is significant and providing the key messages you wish to convey. Please provide the most general claims supported by the evidence and a future perspective on the work.
Reviewer 3 Report
Comments and Suggestions for Authors
Report on the manuscript foods-2657276 entitled: Cell-free supernatant of Lactiplantibacillus plantarum 90: A clean label strategy to improve the shelf life of ground beef gel and its bacteriostatic mechanism.
- L. 80. I am not fond of non-scientific wording. “intuitively” meaning?
- Table 2. Please, review the letter assignment for the last value of Cohesiveness.
- Figures 2 and 3. Error bars have to be reviewed. Sometimes only half of the error bar is shown. In addition, the letter assignment for mean difference should be also included in the Figures.
- Figure 3. Statistics?
- Figure 4. Please, improve the image resolution to able the review.
- Figure 5. Shouldn´t be the SE bars centred? Please, review. Some of the error bars show a longer side.
Or is it possible that the bars have another meaning?
- Doble-check formatting lines: 82, 123, 131, 217, 241, 281.
Comments on the Quality of English Language--
Round 2
Reviewer 1 Report
Comments and Suggestions for Authors
minor revision, detailed comments please see the attachment.

Comments on the Quality of English Languageno
Reviewer 2 Report
Comments and Suggestions for Authors
The authors have addressed all my comments for this paper. Therefore, I have no further comments.
Author Response
Thanks for your comments.
Reviewer 3 Report
Comments and Suggestions for Authors
The authors have carried out a good revision and improvement of the manuscript.
Some minor comments:
- Figures 2, 3 and 4. Letters to separate means are missing. And a clear statistical difference can be observed between some times.
- The format of the bibliography within the text is not correct and some language mistakes can be found but I suppose that the final editing will take care of them.
Comments on the Quality of English Language--
